# Doping with Niobium Nanoparticles as an Approach to Increase the Power Conversion Efficiency of P3HT:PCBM Polymer Solar Cells

**DOI:** 10.3390/ma16062218

**Published:** 2023-03-10

**Authors:** Elmoiz Merghni Mkawi, Yas Al-Hadeethi, Bassim Arkook, Elena Bekyarova

**Affiliations:** 1Department of Physics, Faculty of Science, King Abdulaziz University, Jeddah 21589, Saudi Arabia; 2Center of Nanotechnology, King Abdulaziz University, Jeddah 42806, Saudi Arabia; 3Department of Chemistry, University of California at Riverside, Riverside, CA 92521, USA

**Keywords:** P3HT:PCBM polymer, niobium (Nb) nanoparticles, solar cells, surface morphology

## Abstract

Metal additive processing in polymer: fullerene bulk heterojunction systems is recognized as a viable way for improving polymer photovoltage performance. In this study, the effect of niobium (Nb) metal nanoparticles at concentrations of 2, 4, 6, and 8 mg/mL on poly(3-hexylthiophene-2,5-diyl) (P3HT)-6,6]-phenyl C61-butyric acid methyl ester (PCBM) blends was analyzed. The effect of Nb volume concentration on polymer crystallinity, optical properties, and surface structure of P3HT and PCBM, as well as the enhancement of the performance of P3HT:PC61BM solar cells, are investigated. Absorption of the P3HT:PC61BM mix is seen to have a high intensity and a red shift at 500 nm. The reduction in PL intensity with increasing Nb doping concentrations indicates an increase in PL quenching, suggesting that the domain size of P3HT or conjugation length increases. With a high Nb concentration, crystallinity, material composition, surface roughness, and phase separation are enhanced. Nb enhances PCBM’s solubility in P3HT and decreases the size of amorphous P3HT domains. Based on the J–V characteristics and the optoelectronic study of the thin films, the improvement results from a decreased recombination current, changes in morphology and crystallinity, and an increase in the effective exciton lifespan. At high doping concentrations of Nb nanoparticles, the development of the short-circuit current (J_SC_) is associated with alterations in the crystalline structure of P3HT. The highest-performing glass/ITO/PEDOT:PSS/P3HT:PCBM:Nb/MoO_3_/Au structures have short-circuit current densities (J_SC_) of 16.86 mA/cm^2^, open-circuit voltages (V_OC_) of 466 mV, fill factors (FF) of 65.73%, and power conversion efficiency (µ) of 5.16%.

## 1. Introduction

Organic solar cells (OSCs) have been the subject of intense research interest recently, due to their many desirable qualities, including their portability, low price, simple processing, environmental friendliness, and the possibility of producing them on a large scale using a flexible roll-to-roll method [1,2,3]. Energy is converted in OSCs by the generation of excitons upon light absorption, their subsequent diffusion, their dissociation into free carriers at the junction between the donor and acceptor, and their subsequent transport to the corresponding electrodes [4,5]. Blends of regioregular poly(3-hexylthiophene-2,5-diyl) (P3HT) and fullerene derivative [6,6]-phenyl C61 butyric acid methyl ester (PCBM) are some of the most extensively investigated D-A combinations reported [6,7].

Due to their poor solubility, P3HT chains tend to collect and form crystalline aggregates in mixed solutions. However, the increased aggregate size of P3HT reduces the shelf life of the solution and hinders the mixing of blends [8]. Since the authors report red-shifted absorption and a higher optical absorption coefficient for the additive used to improve device performance, carrier equilibrium mobility and charge carrier mobility were also enhanced [9]. Donors and acceptors often create a phase-separated network, with charge separation occurring at the interfaces. Consequently, morphology plays an additional significant influence in reaching higher efficiency [10]. Recent research indicates that the integration of metallic or semiconducting nanostructures at the transport layers or in the active layer are the most effective light-trapping techniques [11]. Numerous materials have been investigated for use in the active layer, including carbon nanotubes (CNT), silver nanoparticles, graphene, quantum dots, etc. [12,13]. As metallic nanostructures inserted into the active layer may provide local surface plasmon resonance, incoming light with a comparable mode of frequency can be absorbed and/or scattered more effectively because its energy is stored in the oscillation mode of the nanostructures [14].

This nanoparticle (NPs) could be improved further through the application of suitable treatments (e.g., thermal annealing, additives, solvent annealing, etc.) Due to improved interchain interactions, more ordered molecule packing improves charge transport (i.e., better mobility) and absorption at longer wavelengths [15]. However, the shape and size of nanostructures also play a crucial role in light trapping, hence, numerous researchers have described diverse nanostructures such as nanoparticles and nanorods [11,16].

Niobium is a ductile, paramagnetic metal that, at cryogenic temperatures, turns superconducting [17]. In its purest form, it is incredibly soft, but impurities make it tougher. It has a low thermal neutron capture cross-section and a resistivity of 13–16 Ω m at 20 °C. Nanoparticles of niobium are resistant to high temperatures. Niobium is used in electrocatalysts, superconductors, and fuel cells [18].

This research will investigate the first parallel study of high conductivity niobium (Nb) nanoparticles added to a P3HT:PCBM photoactive blend. Due to the strong interchain and interlayer interactions, the Nb-doped compound exhibits a notable redshift and prolonged absorption. Nb doping concentration results in the formation of crystalline domains of P3HT and PCBM, which improves charge collecting effectiveness. In addition, it is observed that the presence of the Nb-doped P3HT:PCBM system substantially alters the nanoscale phase morphology, as revealed by an AFM image. The ability to observe the changes in charge separation, charge transport, and charge collecting is made possible by the viewing of the nanoscale mix morphology. The active layer characteristics and solar cell performance are highly reliant on the Nb doping concentration. The solar cells are respectively standard structure with glass/ITO/PEDOT:PSS/P3HT:PCBM:Nb/MoO_3_/Au exhibit V_OC_ = 466 mV, J_SC_ = 16.86, FF = 65.73%, and a power conversion efficiency (η) of 5.16%.

## 2. Experimental Section

As shown in Figure 1a, an ITO-coated glass substrate with a sheet resistance of 10 Ω^2^ was utilized. ITO was cleaned and agitated in a bath sonicator for 10 min with isopropyl alcohol, acetone, methanol, and de-ionized water. On a clean ITO substrate, a hole transport PEDOT:PSS layer (thickness: 70 nm) was deposited by spin coating at 4000 rpm for 60 s. These samples were annealed at 150 °C for 30 min in the atmosphere. The substrates were transported to a glove box containing N_2_ in order to fabricate the photoactive layer and hole transport PEDOT:PSS layer.

A combination of P3HT and PCBM (20 mg/mL, 1:1 *W/W*) was dissolved in a 15 mg/mL chlorobenzene solvent over 12 h of continuous stirring. To investigate the influence of niobium nanoparticles on the thin-film characteristics of P3HT:PC 61 BM, five solutions were formulated. The solutions contained pure P3HT:PCBM, and different weights of niobium nanoparticles (2, 4, 6, and 8 mg) mixed in 1 mL of chlorobenzene were added to each of the four solutions and stirred thoroughly. Later, the solutions were filtered through 0.45 μm syringe filters. The blended layer was then spin-coated at 2500 rpm for 50 s onto the composite and single PEDOT:PSS layer, followed by 30 min of annealing at 70 °C. Using a thickness profilometer, the thickness was determined to be around 1 μm. The devices were finalized using the e-beam evaporator by depositing a MoO_3_ layer 8 nm thick and then an Au layer 110 nm thick via a shadow mask under a base vacuum of 10^6^ Torr to create an active area of 0.7 cm^2^.The structure of the devices reported here is glass/ITO/PEDOT:PSS/P3HT:PCBM:Nb/MoO_3_/Au.

The structural features of pure P3HT:PCBM and the P3HT:PCBM:Nb thin films were evaluated by X-ray diffraction (XRD, RIGAKU miniflex 600) at = 1.5405 Å radiation. Ultraviolet–visible–near-infrared (UV–vis–NIR) spectrophotometry was used to acquire the blends’ UV–visible spectroscopy ((Varian Cary 100 Bio UV-Visible

Spectrophotometer- Spectralab Scientific, Markham, Ontario, Canada). To examine the optical properties, a Hitachi F-7000 spectrophotometer fitted with a 150 W xenon lamp was used to measure the photoluminescence (PL) emission spectrum at room temperature. Tapping-mode atomic force microscopy (AFM) studies were conducted in an air at ambient temperature using an intermittent contact mode Keysight 5500 scanning probe microscope. Raman spectra were gathered using a Renishaw Raman microscope (= 514 nm). FTIR measurements were conducted using a Nicolet 8700 (Thermo Electron, Waltham, MA, USA) spectrometer for Fourier-transform infrared spectroscopy (FT-IR). A single-layer film’s thickness was determined by a surface profiler (Alpha 500). Current density–voltage (I–V) properties of photovoltaic (PV) cells were determined using a Keithley 236 source meter under AM1.5G illumination from a verified solar simulator emitting 100 mW/cm^2^ of irradiance.

## 3. Results and Discussion

Figure 2 depicts a typical XRD pattern for pure P3HT:PCBM and P3HT:PCBM:Nb blends that have been generated with varying concentrations of Nb nanoparticles (2, 4, 6, and 8 mg/mL). The produced films exhibit two diffraction peaks, which occur at 2θ = 5.49° and 10.81°, and are indexed to the (100) and (200) planes, respectively, which is in good agreement with previous works [6,7].

The first peak is connected to the interchain arrangement in P3HT, which is associated with the interdigitated alkyl chain arrangement [19]. Thus, independent of the presence of fullerene, blended polymers have good crystalline characteristics. The fullerene intersects with the P3HT crystal, however its influence is only observed at the peak density rise. P3HT’s extremely crystalline nature is a result of its self-orientation through interchain assembly [20]. The second discovered peak corresponds to the process known as lamellar stacking of P3HT polymer backbone. These peaks suggest that the P3HT crystallization has increased, which is crucial when the delocalized states are aligned along the transport direction [21,22]. In addition, the results demonstrate that varied doping Nb nanoparticles have a significant impact on the crystallization of the active layer, as demonstrated by the increased peak intensity and the narrower peak shape. Doping could result in a more uniform morphology for P3HT–PCBM blends, with PCBM crystalline aggregates, well-oriented P3HT nanocrystals, and amorphous P3HT–PCBM patches coexisting in a homogeneous, interconnected network.

Using Deby Scherrer’s formula [23], the crystallite sizes of pure P3HT:PCBM and P3HT:PCBM:Nb mix films were determined using the full width at half maximum (FWHM) value of the (100) plane. According to Scherrer’s formula, the mean crystallite size (D) is 15.34, 16.57, 21.55, 27.53, and 31.65 nm for films made with pure and varied Nb nanoparticle concentrations of 2, 4, 6 and 8 mg/mL, respectively, which is in good agreement with previous works [24,25]. Increasing the Nb doping ratios in blended films induces an increase in crystalline size. This finding may be explained by the greater fullerenes’ effect on the crystallinity of the thin films, which corresponds to an increase in their peak intensity.

Using a laser source with an excitation wavelength of 514 nm and a wavenumber range of 500–2000 cm^−1^, Raman spectroscopy was utilized to determine that the clusters are rich PCBM aggregates. Figure 3 displays Raman spectra of pure P3HT:PCBM and the P3HT:PCBM:Nb composite thin films fabricated utilizing varying concentrations of Nb nanoparticles (2, 4, 6, and 8 mg/mL). The principal in-plane ring Raman peaks at 1449 cm^−1^ belong to C=C asymmetric stretching [26], whereas the peak at 1379 cm^−1^ refers to C-C intra-ring stretching vibrations that originate from P3HT [27].

Narrowing of the Raman band width is related to -C=C- stretching, indicating increased crystallinity of P3HT polymer chains [28]. The Raman bands at 1379 and 1449 cm^−1^ for pure P3HT are thought to be sensitive to the π-electron conjugation length of P3HT molecules [29,30]. The Raman band found at around 1206 cm^−1^ corresponds to the inter-ring C-C skeletal stretch mode [31]. At 726 cm^−1^, another Raman band of P3HT is ascribed to the deformation vibration of the C-S-C bond, which is attributed to the deformation vibration of the C-S-C bond [32]. The Raman bands at 1090 cm^−1^ correspond to the C-H bending mode coupled with the C-C interring stretch mode [33].

The main peak seen at 1566 cm^−1^ for PCBM corresponds to the A1′ mode [34]. The immaculate PCBM films have a single prominent peak at 1566 cm^−1^, showing that the molecules are monomeric. By raising the Nb doping concentration, the intensity of these peaks is substantially enhanced. Doping induces the movement of P3HT and the aggregation of PCBM, resulting in extensive phase separation and the rupture of bi-continuous phases. All the Raman scattering signals of the plasmonic samples are amplified, which is consistent with the increase in extinction spectra. For P3HT:PCBM:Nb films, peaks corresponding to both P3HT and PCBM’s dominant peak are found, suggesting the existence of both P3HT and PCBM. Raman spectra obtained from thin P3HT:PCBM:Nb films appear to be similar and unshifted. This demonstrates the great uniformity of the active layer and may be attributable to the higher crystallinity of P3HT in the active layer.

FT-IR spectroscopy was utilized to establish that functionalization had occurred satisfactorily. Figure 4 displays the FTIR spectra of pure P3HT:PCBM and P3HT:PCBM:Nb blends created by doping Nb nanoparticles in various concentrations. The detected FTIR band at 735 cm^−1^ matches to methyl rock [35]. The band found at 821 cm^−1^ corroborates the existence of aromatic C-H outside of simple vibrations and suggests that the structure of P3HT is composed of P3HTchains [36]. The peak at 1107 cm^−1^ is ascribed to the asymmetries of the C-O stretching bands [37]. The carbon-to-carbon single bond (C-C) is represented by the band at 1270 cm^−1^ [38]. The carbon-to-carbon double bond (C=C) symmetric ring stretching is responsible for the band seen at 1454 cm^−1^ [39]. We detected an asymptotic methyl stretching band at 2960 cm^−1^ [39,40].

A carbonyl functional group peak is observed at 1728 cm^−1^, which corresponds to the carbonyl of the lactone ring [41]. C-C stretching of the benzene ring is responsible for the 1598 cm^−1^ peak [42]. As with the 1:1 P3HT:C60 film, a new band forms at approximately 2360 cm^−1^ [43]. These bands become more intense as the amount of Nb nanoparticles increases. This 2360 cm^−1^ band is not due to C60, consequently, this variation in vibrational energy may be a result of the charge transfer between the sulfur atom in the P3HT molecule and C60. As seen in Figure 4, the strength of conspicuous bands rises as incoming ion fluences increase. The existence of all modes of P3HT indicates that the fundamental unit structure of the P3HT molecule is not appreciably affected by low-fluence ion irradiation. Self-stacking of thiophene rings among adjacent polymer chains may result in chain rearrangement due to ion irradiation at the fluence for regioregular P3HT, which comprises of self-packing of the hexyl side chains [44].

We studied and investigated polymer surface morphology using an atomic force microscope (AFM) in air. Figure 5 displays an AFM picture of virgin P3HT:PCBM:Nb thin films produced by doping Nb nanoparticles in various concentrations. The surface P3HT:PCBM:Nb (2 mg/mL) layer is rather rough with a tiny nodule-like structure, indicating a broad extent of continuous domains with a root mean square (rms) roughness of 9.78 nm (Figure 5a). As demonstrated in Figure 5b, the P3HT:PCBM Nb (4 mg/mL) layer is somewhat smoother (rms roughness = 7.93 nm) and seems homogeneous. The P3HT:PCBM:Nb (6 mg/mL) exhibits a somewhat smooth surface (rms roughness = 5.67) as seen in Figure 5c. Upon raising the Nb concentration to 8 mg/mL (Figure 5c), the surface (rms roughness = 3.51) becomes less noticeable as both length and width decrease, indicating that the surface roughness decreased, demonstrating that the loading of PCBM affects the P3HT-b-PS self-assembly behavior.

Large aggregates emerge on the surface, most likely due to the self-assembly of the di-block copolymer. Therefore, as the concentration of Nb increases, the characteristic length scale in the films grows and the matching structure becomes well-defined. It is widely recognized that both microphase separation and P3HT crystallization influence the morphological organization of rod block polymers including a P3HT section. Large aggregates are seen in a film with a low concentration of Nb doping, and these aggregates may limit light absorption in the aggregated areas [45]. It is likely that the heat treatment and Nb doping allow PCBM particles to quickly agglomerate inside the soft P3HT, resulting in extensive phase separation and a substantial increase in the roughness of the P3HT:PCBM film. In addition, heterogeneous nucleation of the P3HT crystallites occurs at the substrate interface, leading to a considerable reorientation of the P3HT crystallites. The P3HT lamellae and π–π stacking direction are preferentially aligned parallel to the substrate in melt-crystallized P3HT [46]. We observe that the exposure of the P3HT:PCBM layer to Nb doping considerably alters the layer’s shape. The most abrasive surface may accelerate hole collection and so contribute to the enhancement of J_SC_. These observations imply that the surface modifications in the P3HT:PCBM layer after doping are typically attributable to the decreased production of PCBM aggregates and not to alterations in the crystalline ordering of the P3HT chains [47].

The schematic energy level diagram of the energy and charge transfer effects of P3HT:PCBM:Nb blends is depicted in Figure 6. In the electrical model, Nb nanoparticles may serve as locations for hole hopping, electron scattering, and carrier recombination. As illustrated in Figure 6, it is believed that the device consists of a single semiconductor with the lowest unoccupied molecular orbital (LUMO) of the acceptor and the highest occupied molecular orbital (HOMO) of the donor serving as the valence and conduction bands, respectively. Under the influence of light, a large number of photogenerated excitons, at the interface of P3HT and PCBM, disintegrate into free electrons and holes. Additionally, the energy steps between the highest (HOMO) levels of P3HT and Au and the lowest (LUMO) levels of PCBM and Au make it easier to transport carriers and collect carriers. Thus, there are three potential electron transfer pathways in the P3HT:PCBM:Nb system: electron transfer from P3HT to PCBM; electron transfer from P3HT to MoO_3_; and electron transfer from P3HT to Nb and, subsequently, to PCBM. Thus, the incorporation of Nb might lead to a more efficient photo-induced charge transfer, i.e., a more efficient dissociation of photogenerated excitons.

The quenching of photoluminescence in a bulk heterojunction is a useful measure of the efficacy of exciton dissociation and the charge transfer efficiency of donor–acceptor composite materials. Figure 7 displays the photoluminescence spectra of pure P3HT:PCBM and P3HT:PCBM:Nb thin films at room temperature synthesized by doping Nb nanoparticles with a variety of dopants. The PL has five unique characteristics at 584 nm, 684 nm, 731 nm, 778 nm, and 897 nm. Compared with the pristine P3HT:PCBM, the PL intensities of P3HT:PCBM:Nb thin films fabricated by doping Nb NPs are significantly quenched and show a red shift. The red shift of emission peaks located at 644 nm to 653 nm corresponds to band-to-band emission in P3HT, which has an energy gap of roughly 1.9 eV [48,49]. The PL quenching is dramatically enhanced after Nb doping; in particular, a thin film prepared using doped Nb nanoparticles at 8 mg/mL is the most beneficial for quenching photo-induced photons on the active layer. The quenching of the peak in PL intensity in the P3HT:PCBM:Nb thin films can be related to charge transfer and the improved exciton separation efficiency, which are consistent with earlier findings in the scientific literature [50,51].

The photoluminescence quenching of P3HT:PCBM:Nb may be attributable to the π–π interaction of P3HT with Nb nanoparticles, which creates extra decaying routes for excited electrons inside the Nb nanoparticles. The AFM pictures reveal that the phase-separated crystalline P3HT:PCBM:Nb films are more evenly dispersed with a percolated network. This conclusion correlates nicely with the issue of exciton separation. Thus, excitons generated within the randomly distributed big aggregates in the P3HT:PCBM:Nb film undergo severe recombination, despite the fact that each large aggregate aids in charge transfer, resulting in a modest improvement in J_SC_ and PCE.

The UV–vis absorption spectra of pure P3HT:PCBM and P3HT:PCBM:Nb synthesized and annealed at 150 °C with various Nb concentrations in the solvent are displayed in Figure 8. For the blend with a modest concentration of Nb nanoparticles (2 and 4 mg/mL), the peak absorption wavelength of the film is 503 nm, with hardly discernible shoulders at 551 nm and 603 nm. With a high Nb content (6 and 8 mg/mL) in the P3HT:PCBM mixture, the highest peak absorption at 517 nm is more intense, and the shoulders at 557 nm and 607 nm are more pronounced. The existence of discrete vibronic absorption shoulders (at 557 nm and 607 nm) accounts for the increased conjugation duration and more organized structure of P3HT [21,52]. At increasing Nb concentrations, however, this shoulder tends to become smoother. This indicates that the connection between P3HT polymer chains and PCBM molecules has been disrupted.

Overall, we believe the improvement in absorption is mostly attributable to the diffusion of PCBM and development of needle-like crystals that cover the substrate well (AFM data), followed by cluster formation at increasing Nb nanoparticle concentration [24]. Changes in polymer film absorption spectra are consistent with the greater peak intensities of XRD. These changes in polymer film absorption spectra can be explained by the influence of the P3HT molecule’s π–π* transition, which is connected to the augmentation of absorption intensity [53]. This improved absorption might be ascribed to the greater interchain interaction between P3HT chains because of more delocalized conjugated electrons, which results in a narrower band gap between π and π * [54]. This results in an increase in the optical π–π * transition and a redshift in the peak absorption wavelength. The increase in effective conjugation length in π–π* conjugated systems adds to a reduction in the energy gap of the composite thin film, allowing for the production of a greater number of excitons, which leads to a greater photocurrent.

The photocurrent density–voltage curves of organic solar cells devices in the configuration of glass/ITO/PEDOT:PSS/P3HT:PCBM:Nb/MoO_3_/Au produced utilizing organic pure P3HT:PCBM and P3HT:PCBM:Nb active layer in varied Nb nanoparticles concentrations are shown in Figure 9. Table 1 lists the fundamental characteristics of the developed organic solar cells, including their short-circuit current density (J_SC_), open-circuit voltage (V_OC_), fill factor (FF), power conversion efficiency (PCE-η), series resistance (Rs), and shunt resistance (Rsh). The best P3HT:PCBM:Nb device prepared with 8 mg/mL-Nb nanoparticles exhibits a short-circuit current density (J_SC_) of 16.86 mA/cm^2^, an open-circuit voltage (V_OC_) of 466 mV, a fill factor (FF) of 65.73%, and a power conversion efficiency (η) of 5.14% under AM 1.5 G1 sun illumination (100 mW/cm^2^). According to these findings, the P3HT:PCBM:Nb-based device exhibits photovoltaic capabilities in the presence of light, and when the Nb concentration in the active layer grows, the device’s J_SC_ and PCE increase, along with a little drop in V_OC_.

The improvement in crystallization of the P3HT:PCBM:Nb organic semiconductor with increased Nb doping may account for this rise. In addition, low PCBM degradation may result in less load carrier deterioration, which can raise the PCE of the device [30]. In addition, an examination of the UV–vis peaks in the absorption spectrum (see Figure 8) reveals an increase in light absorption, which results in a higher current. Based on the rise in rms seen in the AFM images (Figure 5), there is an increase in the contact area between the polymer film and the buffer layer and gold cathode, resulting in a decrease in the series resistance from 25.64 to 15.4 Ω cm^2^ (Table 1). After doping, the film surface roughness becomes smoother, and, as a result, Rs is seen to drop. In addition to this impact, the addition of Nb to the active layer boosts the load-bearing capability of the device, resulting in an increase in the J_SC_ of the device. According to Table 1, as the Nb concentration in the active layer grows, the device’s shunt resistance climbs. On the other hand, the inclusion of Nb nanoparticles in the structure of the device lowers the contact resistance between the metal and the active layer, resulting in a reduction in the device’s series resistance. By minimizing carrier aggregation in the interface, decreasing the series resistance boosts the current density and power conversion efficiency of the device [55,56].

## 4. Conclusions

We have attempted to develop a spin-coating method for fabricating P3HT:PCBM-based organic solar cell devices with varying concentrations of Nb nanoparticles. Several intriguing impacts of Nb doping concentration on optical, structural, and electrical characteristics have been observed, which we feel are essential for improving device PCEs. Absorption spectra of P3HT:PCBM:Nb thin film reveal three emission peaks at 517 nm and more distinct shoulders at 557 nm and 607 nm, which are attributed to crystalline aggregates of P3HT with a longer-type shorter π-stacking distance, resulting in the observation of a higher and lower energy emitter, respectively. Increasing the amount of Nb in a composite P3HT:PCBM:Nb thin film favors the creation of domains with near π–π stacking between thiophene units. In this instance, we also observe the formation of stacked nodule-like structures by both microphase separation and P3HT crystallization. The addition of Nb to the mixture enhances the P3HT–PCBM phase separation, which, in turn, impacts the exciton dissociation efficiency. These characteristics have a direct impact on the photovoltaic qualities. The structure of glass/ITO/PEDOT:PSS/P3HT:PCBM:Nb/MoO_3_/Au fabricated from the P3HT:PCBM:Nb active layer synthesized with 8 mg/mL Nb has the best efficiency as (η) of 5.16% with (J_SC_) of 16.86 mA/cm^2^, (V_OC_) of 466 mV, and (FF) of 65.73%. With these findings, this study reveals that a suitable application of Nb nanoparticles is an excellent way to manage film morphology, absorbance, and crystallinity, and, hence, enhance the photovoltaic performance of OPV devices.

## Figures and Tables

**Figure 1 materials-16-02218-f001:**
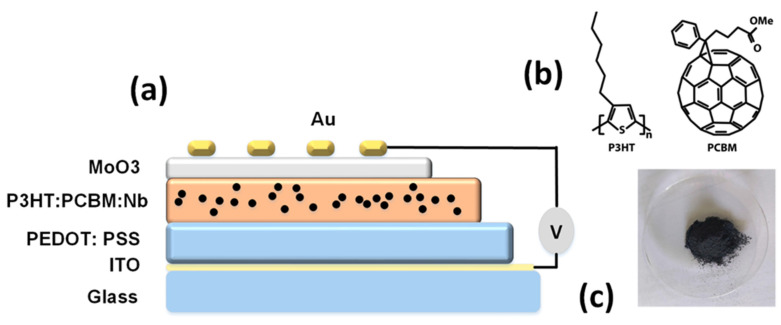
(**a**) Solar cell with a glass/ITO/PEDOT:PSS/P3HT:PCBM:Nb/MoO_3_/Au structure with Nb nanoparticles embedded in active layers, (**b**) molecular structures of P3HT and PCBM, and (**c**) niobium (Nb) nanoparticles.

**Figure 2 materials-16-02218-f002:**
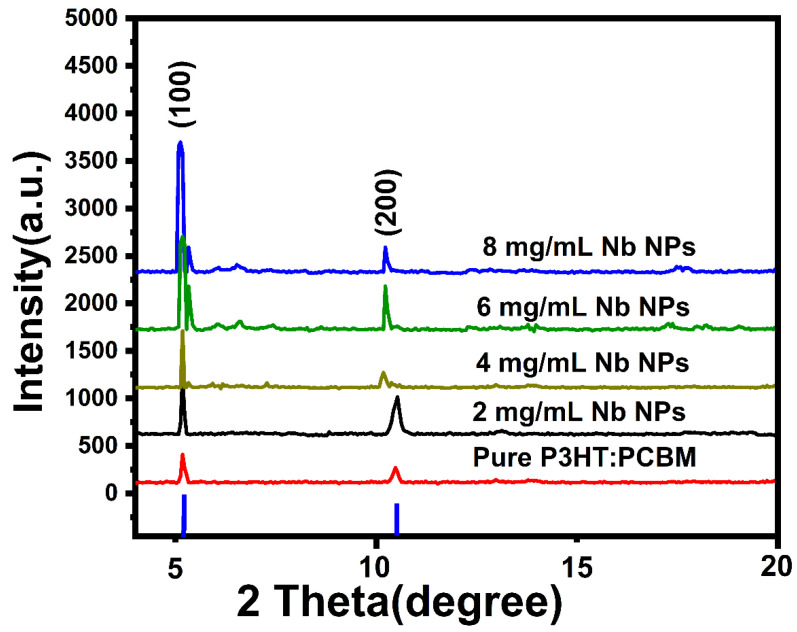
XRD patterns for pure P3HT:PCBM and P3HT:PCBM:Nb films fabricated using different Nb nanoparticle concentrations.

**Figure 3 materials-16-02218-f003:**
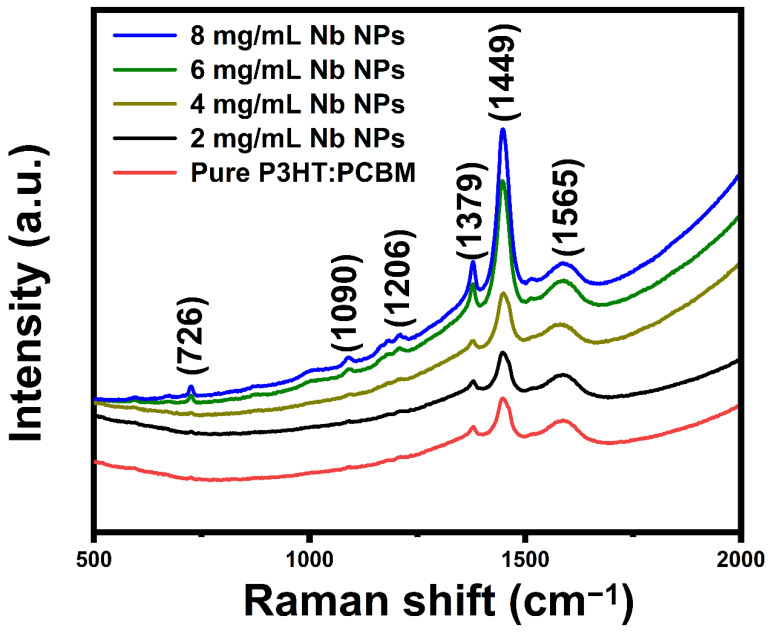
Raman spectra of pure P3HT:PCBM and P3HT:PCBM:Nb thin films synthesized with varying amounts of Nb nanoparticles.

**Figure 4 materials-16-02218-f004:**
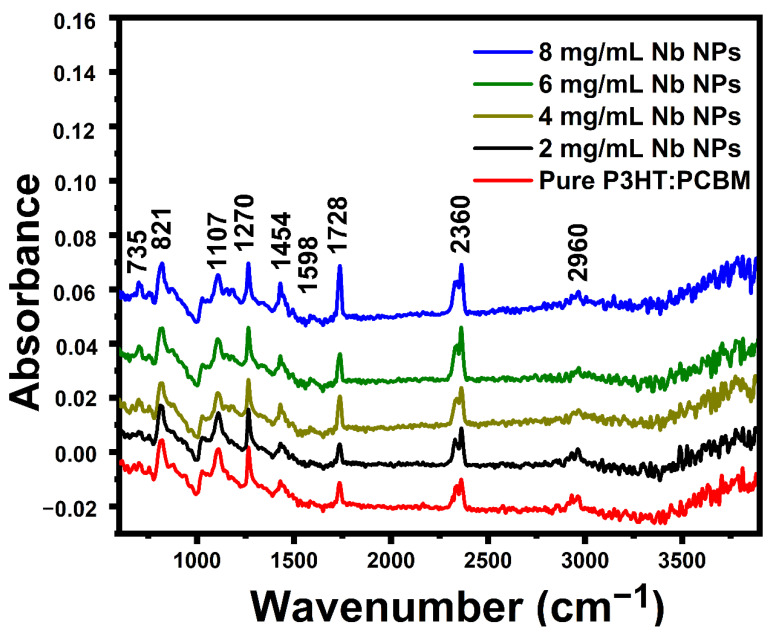
FT-IR spectra of pure P3HT:PCBM and P3HT:PCBM:Nb blends synthesized with various Nb nanoparticle doping concentrations.

**Figure 5 materials-16-02218-f005:**
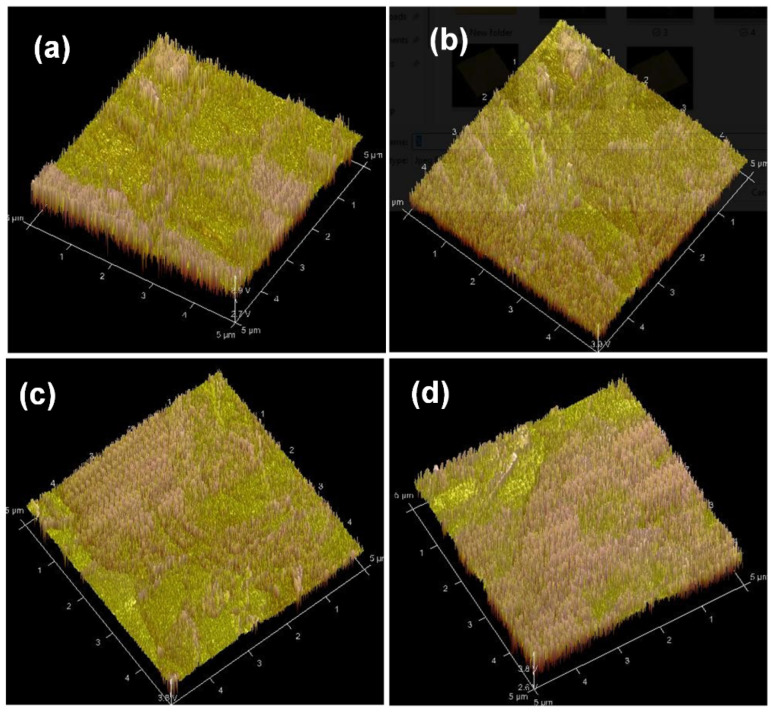
Atomic force microscopy (AFM) topographic images revealing major structural differences at the surface of P3HT:PCBM:Nb photoactive layers prepared using different doping Nb nanoparticles of (**a**) 2, (**b**) 4, (**c**) 6, and (**d**) 8 mg/mL.

**Figure 6 materials-16-02218-f006:**
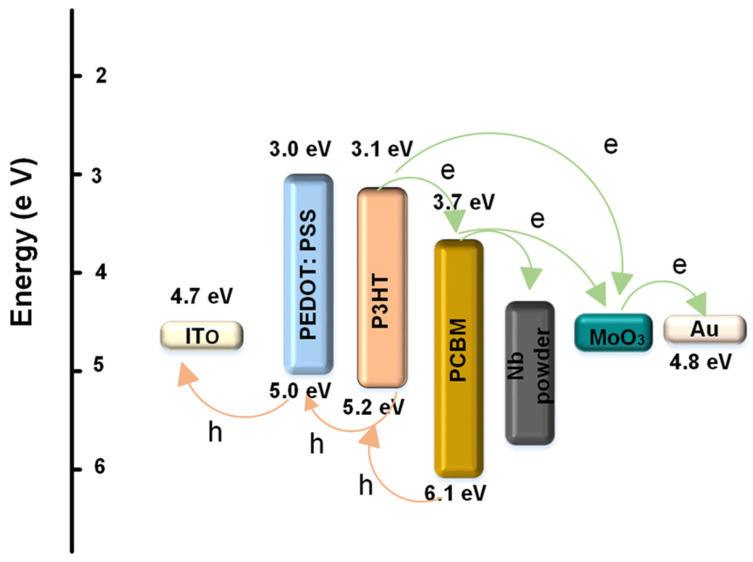
Diagram of energy levels for a P3HT PCBM:Nb bulk heterojunction solar cell with an interface layer.

**Figure 7 materials-16-02218-f007:**
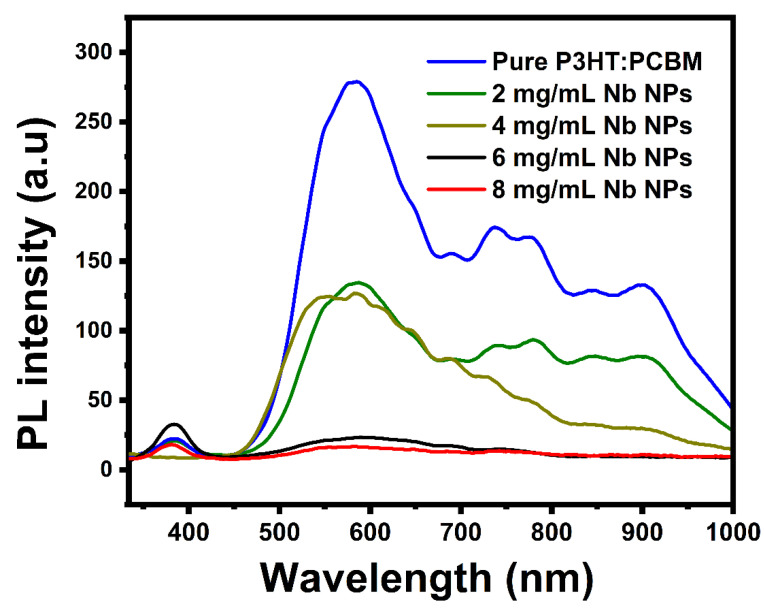
Photoluminescence (PL) spectra of pure P3HT:PCBM and P3HT:PCBM:Nb thin films produced with different amounts of Nb nanoparticles.

**Figure 8 materials-16-02218-f008:**
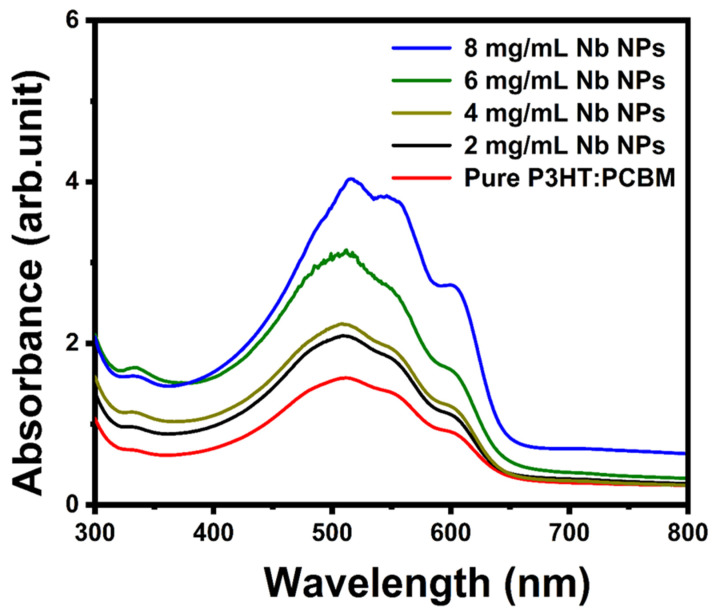
Optical absorption spectra of pure P3HT:PCBM and PCBM:P3HT thin films doped using different Nb concentrations.

**Figure 9 materials-16-02218-f009:**
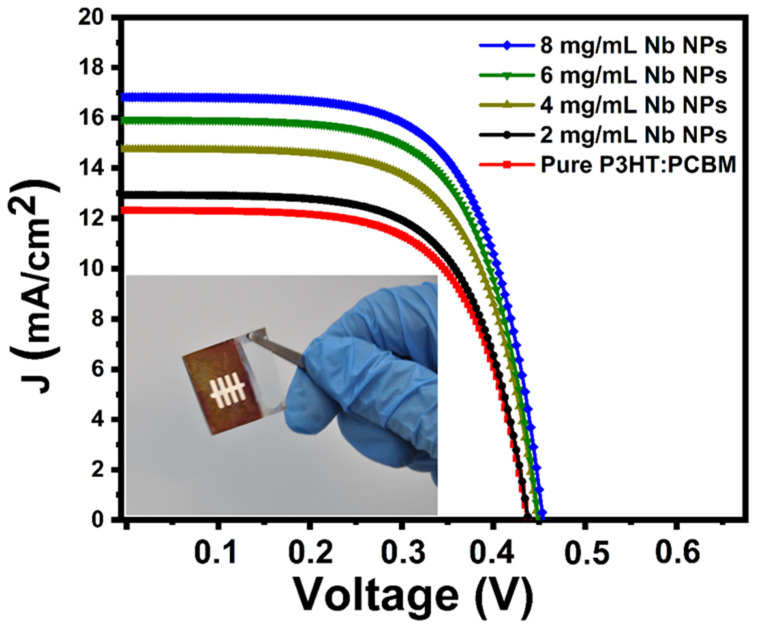
J–V plots of the glass/ITO/PEDOT:PSS/P3HT:PCBM:Nb/MoO_3_/Au solar cell device based on an active layer P3HT:PCBM:Nb and pure P3HT:PCBM.

**Table 1 materials-16-02218-t001:** Photovoltaic performance of pure P3HT:PCBM and the P3HT:PCBM:Nb solar devices were fabricated utilizing various Nb concentrations under AM 1.5 sun irradiation with 100 mW/cm^2^ of light power.

Sample	V_OC_ (mV)	J_SC_ (mA/cm^2^)	FF (%)	η (%)	$\mathrm{Rs}\;(Ω\;cm^{2})$	$R_{\mathrm{sh}}\;(Ω\;cm^{2})$
**Pure P3HT:PCBM**	445	12.3	62.42	3.41	26.87	412.5
**2 mg/mL-Nb**	445	12.95	62.53	3.6	25.64	453.3
**4 mg/mL-Nb**	455	14.74	63.73	4.27	22.34	625.1
**6 mg/mL-Nb**	458	16.01	64.23	4.7	17.23	724.1
**8 mg/mL-Nb**	466	16.86	65.73	5.16	15.67	944.2

## Data Availability

The data presented in this study are available from the corresponding author, Elmoiz Mkawi, upon reasonable request.

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
