# Peer review of "Doping with Niobium Nanoparticles as an Approach to Increase the Power Conversion Efficiency of P3HT:PCBM Polymer Solar Cells"

_materials, 2023, doi:10.3390/ma16062218_

Round 1

Reviewer 1 Report

1. The author add niobium nanoparticles as the third component in to P3HT:PCBM to increase the performance of the solar cells. However, the recent strategy of incoporation of third component such as donors or acceptors are not covered in the introductions. Since the author discuss on the charge carrier transportation and charge transfer in the paper, the following paper should be cited. Nano Lett. 2019, 19, 8, 5053–5061

2. Page 3 line 99, the authors mention used 80nm thick MoO3 for electrodes. People usually used less than 10 nm MoO3 as the interfacial layer and then deposit Au or Ag. The 80 nm is way more thicker than the common sense. Thick MoO3 layer would cause high resistance to the device. Can the authors double check about the thickness of MoO3?

3. The AFM image in Figure 5 looks not very homogenous as the light color distribution is not well cover. Can the author identify the Nb region? If possible can use Ramen mapping?

4. Since the HOMO of Nb is higher than the PCBM, can the authors explain how the holes can transfer from Nb to PCBM?

5. Both Figure 9 and Table 1 should include the P3HT:PCBM as the reference so would be able to make comparison.

Reviewer 2 Report

The paper is a part of series of articles about doping of P3HT:PCBM polymer solar cells with nanoparticles (10.1016/j.jlumin.2021.118420; 10.3390/polym13132152). All of that articles have one scheme, which is a origin of some mistakes.

Some of them:

1)      (line 99) The devices were finalized using the e-beam evaporator by depositing a MoO3 layer 80 nm thick and then an Au layer 110 nm thick via a shadow mask under a base vacuum of 106 Torr to create an active area of 0.7 cm2. But on Figure 1(a) you have an Ag.

2)      Figure 3 – you change the amount of Nb NPs (in the text you have different numbers)

3)      Figure 4 – you change the amount of Nb NPs (in the text you have different numbers)

4)      Figure 6 – Nb has a energy bandgap?

In this article authors studying influence of Niobium metal nanoparticles in different concentration for power conversion efficiency. I think that main disadvantage of article is lack of comparison with sample without Niobium particles (Figure 2,3,4,5,7,8,9).

Some, I think, important things:

A)      On Figure 2 you have intensity -500 – what for? You write only about two signals, but on spectra we can see some more signals. What with them? Your signal about 50 get new arm from right side – why it’s looking like that? From which peak crystallite sizes was calculated?

(line 138-140) In addition, the results demonstrate that varied doping Nb nanoparticles have a significant impact on the crystallization of the active layer, as demonstrated by the increased peak intensity and the narrower peak shape. – it’s not true, I can’t see it on your XRD spectra.

B)      (line 166-169) At 726 cm-1 another Raman band of P3HT is ascribed to the deformation vibration of the C–S–C bond, which is attributed to the deformation vibration of the C–S–C bond [33]. The Raman bands at 1090 cm-1 correspond to the C–H bending mode coupled with the C–C interring stretch mode[34]. – why this peaks showed in samples with higher amount of niobium, not earlier?

C)      On Figure 4 you have Absorbance (no Transmittance).

(line 184-185) FT-IR spectroscopy was utilized to establish that functionalization had occurred satisfactorily. How we can see it, when we can’t compare it with sample without niobium? In this description we have to much linkers – all that information we can find in one or two linkers cited in this text.

(line 194-195) A carbonyl functional group peak was observed at 1728 cm-1, which corresponds to the carbonyl of the lactone ring [41]. Why it showing only with higher amount of Niobium?

D)      (line 253-255) Thus, there are three potential electron transfer pathways in P3HT: PCBM: Nb system: electron transfer from P3HT to PCBM; electron transfer from P3HT to Nb; and electron transfer from P3HT to Nb and subsequently to PCBM. On Figure 6 you have direct arrow from P3HT to MoO3, but you don’t mention about it in your text.

E)      (line 267-268) The PL has three unique characteristics at 653 nm, 674 nm, and 742 nm. – What with the other signals?

F)       (line 295-296) The existence of discrete vibronic absorption shoulders accounts for the increased conjugation duration and more organized structure of P3HT – where is this shoulder?

G)     (line 304-305) This can be explained by the influence of the P3HT molecule's ππ transition, which is connected to the augmentation of absorption intensity [54]. – what can be explained?

H)     (line 311-312) A smaller band gap enables the thin film to absorb more photons at lower energies, generating a greater photocurrent. -???

Some little incorrect things:

a)        (line 67-68) Niobium used in electrocatalysts, superconductors, and fuel cells utilize niobium. – to much nobium in this sentence

b)      (line 109) Tapping-mode atomic force microscopy (AFM) studies were conducted in air at ambient temperature using an intermittent contact mode Keysight 5500 scanning probe microscope. And (line 210 - 211) We studied and investigated polymer surface morphology using an atomic force microscope (AFM) in a nitrogen-free environment.

c)       In article you use name “P3HT”, without indexes, on Figure 1,6 you have 3 in index.

d)      Figure 5 – I can’t see the scale of images

e)      Figure 6 – on the left axis – energy is enough

f)        (Line 255-257): In a cell with an inverted structure, electrons are carried to an ITO substrate while holes are sent to an Au electrode, reducing the probability of recombination. I don’t understand the sense of this sentence in this place. It is not connected with Figure 6 or something else.

g)       (line 327): a power conversion efficiency (η) of 4.16 % - in table you have 5,16 %

h)      (line 339): series resistance from 25.64 to 15.4 cm2 - units

Reviewer 3 Report

The manuscript , “Doping with niobium nanoparticles as an approach to in-2 crease the power conversion efficiency of P3HT:PCBM polymer solar cells” by E. M. Mkawi et al., is generally well written and easy to follow.

However, I would like the authors to address the follow comments. In my opinion, without the pristine P3HT/PCBM data, it is difficult to validate the authors’ claims.

1.       For Figure 2, Figure 3, Figure 7 and 8, please add the spectra for just P3HT/PCBM film (no Nb) at the same preparation condition. And provide the device performance for P3HT/PCBM in Figure 9 as the standard.

2.       The color selection for each concentration of Nb nanoparticle should remain to be the same color to be more consistent in Figure 2,3,4,7,8,9.

Round 2

Reviewer 3 Report

 The manuscript on “Doping with niobium nanoparticles as an approach to in-2 crease the power conversion efficiency of P3HT:PCBM polymer solar cells” by E. M. Mkawi et al. has been revised to address my previous comments. Thank you.

Scientifically, I would like the authors to double check their data in Figure 7 because they don’t make sense to me. With the addition of Nb, the rise of photoluminescent intensity at 4 to 8 mg/ml generally indicates an enhanced exciton recombination process, which leads to reduced charge separation and lower device efficiencies. This is contrary to what they observed. The manuscript needs to justify their claim “The rise in PL intensity in the of the P3HT:PCBM:Nb thin films can be related to charge transfer and the improved exciton separation efficiency”. Work in reference 50 proved the opposite, unless the PL intensity is quenched, charge transfer is not efficient and life-time of the charges is shorter. The further explanation in the following paragraph in the manuscript from line 274 to line 285 also doesn’t make sense to me as well. Reference 7 studied the domain sizes of a similar system, however, their work won’t suggest “the enhanced diffusion of PCBM into a highly crystalline P3HT matrix, which leads to an increase in charge transfer and a consequently high PL intensity”.

Author Response

Response to Reviewers

Manuscript ID: materials-2207521

Title: Doping with niobium nanoparticles as an approach to increase the power conversion efficiency of P3HT:PCBM polymer solar cells

Journal of Materials

First of all, in my capacity as the corresponding author, I would like to express my sincere gratitude and appreciation to the reviewer for his/her valuable comments, without which this manuscript would not have been valuable. I am really greatly indebted to the reviewer and to the editor.

Comments made by reviewer #3

Scientifically, I would like the authors to double check their data in Figure 7 because they don’t make sense to me. With the addition of Nb, the rise of photoluminescent intensity at 4 to 8 mg/ml generally indicates an enhanced exciton recombination process, which leads to reduced charge separation and lower device efficiencies. This is contrary to what they observed. The manuscript needs to justify their claim “The rise in PL intensity in the of the P3HT:PCBM:Nb thin films can be related to charge transfer and the improved exciton separation efficiency”. Work in reference 50 proved the opposite, unless the PL intensity is quenched, charge transfer is not efficient and life-time of the charges is shorter. The further explanation in the following paragraph in the manuscript from line 274 to line 285 also doesn’t make sense to me as well. Reference 7 studied the domain sizes of a similar system, however, their work won’t suggest “the enhanced diffusion of PCBM into a highly crystalline P3HT matrix, which leads to an increase in charge transfer and a consequently high PL intensity”.

A1  #: The reviewer is correct, and we deeply apologize for this mistake due to wrong sample numbering during PL analysis. We corrected that information and added new information to the revised manuscript. Please see page 9, line 267 and figure 7. Your suggestions were extremely beneficial to us. Please accept our deep appreciation.